Natural hazards; mental health; fire; drought; vulnerability

**Corresponding author:**
Jane Fisher;
Email: jane.fisher@monash.edu

K.V., S.M. and R.N.K. authors indicate co-first authorship and equal contribution to the manuscript.

# Mental health of vulnerable groups experiencing a drought or bushfire: A systematic review

Karan Varshney[1], Shelly Makleff[1] , Revathi N. Krishna[1], Lorena Romero[2], Julie Willems[1], Rebecca Wickes[3] and Jane Fisher[1] 

[1]Global and Women's Health Unit, Public Health and Preventive Medicine, Monash University, Melbourne, VIC, Australia; [2]Ian Potter Library, Alfred Health, Melbourne, VIC, Australia and [3]School of Criminology and Criminal Justice, Griffith University, Brisbane, VIC, Australia

## Abstract

Natural hazards are increasing because of climate change, and they disproportionately affect vulnerable populations. Prior reviews of the mental health consequences of natural hazard events have not focused on the particular experiences of vulnerable groups. Based on the expected increase in fires and droughts in the coming years, the aim of this systematic review is to synthesize the global evidence about the mental health of vulnerable populations after experiencing natural hazards. We searched databases such as Ovid MEDLINE, EMBASE, CINAHL and Ovid PsycInfo using a systematic strategy, which yielded 3,401 publications. We identified 18 eligible studies conducted in five different countries with 15,959 participants. The most common vulnerabilities were living in a rural area, occupying a low socioeconomic position, being a member of an ethnic minority and having a medical condition. Common experiences reported by vulnerable individuals affected by drought included worry, hopelessness, isolation and suicidal thoughts and behaviors. Those affected by fire reported experiencing posttraumatic stress disorder (PTSD) and anger. These mental health problems exacerbated existing health and socioeconomic challenges. The evidence base about mental health in vulnerable communities affected by natural hazards can be improved by including standardized measures and comparison groups, examining the role of intersectional vulnerabilities, and disaggregating data routinely to allow for analyses of the particular experiences of vulnerable communities. Such efforts will help ensure that programs are informed by an understanding of the unique needs of these communities.

## Impact statement

This systematic review provides synthesized evidence about the mental health of vulnerable populations who have experienced fire or drought. Experiences of depression, posttraumatic stress disorder, anxiety and anger were common, influenced by limited access to mental health services, loss of community and loss of income. Few publications disaggregate data for vulnerable individuals, and multiple measures of mental health are used in this body of literature, limiting our understanding of how the mental health consequences of natural hazards intersect with different forms of vulnerability. Nevertheless, the data indicate that the mental health needs of vulnerable members of the community warrant specific consideration following natural hazards.

## Introduction

Natural hazards are intensifying globally, and climate change is a major contributor to this (WMO, 2021). Natural hazards have great economic costs (Kousky, 2014), and affected populations can experience negative health consequences (Noji, 2000). Drought and wildfires/bushfires (hereafter, fires), which often co-occur, are among the natural hazards expected to increase in frequency and intensity in the coming years. It is predicted that there will be a global increase in extreme fires of 14% by 2030, 30% by the end of 2050 and 50% by the end of the current century (United Nations Environment Programme, 2022). Similarly, by 2100, economic losses due to drought may become five times higher than current levels (European Commission Joint Research Centre et al., 2020).

In addition to their significant economic, social, environmental and political impacts (Middlemann, 2007), fire and drought can have major impacts on the health and well-being of the people who are directly affected by them (World Health Organization, 2022*a*,*b*). In an early review, Laugharne and colleagues found that people directly affected by fire, as well as their close family, are at an increased risk of adverse psychological effects, including

traumatic stress and depression (Laugharne et al., 2011). Exposure to fires is associated with lasting psychological impacts including depression, posttraumatic stress, suicidality and increased drug and alcohol use (McFarlane et al., 1997; Finlay et al., 2012). Similarly, experiencing drought negatively influences mental health in complex and diverse ways (Vins et al., 2015), and is implicated in contributing to mental distress and suicidality (Austin et al., 2018). A recent analysis of the Australian Rural Mental Health Study, a longitudinal study of 1,800 households across rural and remote New South Wales that examines the determinants of mental health as influenced by individual, family and community factors, suggests that while mental distress might abate after about three years of drought exposure, general life satisfaction and ability to maintain good health can continue to decline over time (Luong et al., 2021).

Certain populations have social or physical vulnerabilities that contribute to poor health and well-being and have implications disaster context (Tierney, 2006; Blaikie et al., 2014). Building on the definition by Waisel (2013), for the purpose of this review we define vulnerable populations as including people who are members of ethnic minority groups, are at least 60 years as defined by the World Health Organization, occupy a low socioeconomic position, have a chronic medical condition, are bereaved of a spouse, or reside in rural/remote areas (Waisel, 2013). There is substantial evidence from diverse settings indicating that members of vulnerable groups are at an elevated risk of poor physical health outcomes after experiencing fire/drought (Stanke et al., 2013; Kondo et al., 2019; Walter et al., 2020; Haikerwal et al., 2021). For example, a study focusing on drought mortality from 2000 to 2019 in Brazil showed that excess mortality risk attributable to extreme drought exposure was 0.99%; however, it increased to 2.28% for children, 1.57% in the elderly and 3.19% in women aged 65–74 years – showing that these vulnerable groups had an elevated risk of mortality compared to non-vulnerable groups (Salvador et al., 2022).

While the physical health impacts of fire/drought on vulnerable groups are well documented, less is known about the mental health challenges faced by these populations after natural hazards. Further, the mental health impact of hazards cannot be fully understood when examined in isolation from other individual and social factors (Weldon, 2008) that influence mental well-being. Thus, an understanding of the distinct factors influencing the mental health of vulnerable groups after experiencing fire/drought can inform emerging research priorities and is important for the development of effective interventions to support recovery. In consideration of the increasing threat posed by drought and fire globally, the aim of this systematic review is to describe the global literature examining the mental health of vulnerable populations after experiences of drought or fire and to identify knowledge gaps.

## Methods

### Database searches

This systematic review followed the 'Preferred Items for Systematic Review and Meta-Analyses' (PRISMA-) guidelines (Page et al., 2021). On November 19, 2021, searches were conducted in four different databases: Ovid MEDLINE (Medical Literature Analysis and Retrieval System Online), EMBASE (Excerpta Medica Database), CINAHL (Cumulative Index to Nursing and Allied Health Literature) and Ovid PsycInfo (APA PsycINFO). The

search was repeated on March 22, 2022, to identify the most updated literature. The OSF registered protocol (10.17605/OSF.IO/SQEMC) and full protocol (Makleff et al., 2022) describe the methodology in detail. This study was initially registered as a scoping review based on published guidance (Munn et al., 2018). However, based on the robustness of our methods that fulfill the PRISMA guidelines (Page et al., 2021), including quality appraisal by two researchers, and a precise research question, this review is more appropriately described as a systematic review and is presented as such in this paper.

We focused on bushfire and drought in this review because these forms of natural hazard are becoming increasingly common in geographic regions across the globe due to similar reasons, such as high temperatures, low humidity and strong winds, may co-occur, are exacerbated by escalating climate change, and have significant socioeconomic and health impacts (Middlemann, 2007; Richardson et al., 2022; World Health Organization, 2022a,b). The search strategy utilized a combination of database-specific subject headings and free text terms that cover three concept areas: (a) bushfires, wildfires and natural disasters; (b) mental health and well-being; and (c) disadvantaged and vulnerable populations. Search terms were inclusive to cover qualitative approaches including grounded theory, focus groups, phenomenology and interviews; and quantitative methodologies including cohort designs, cross-sectional studies and case–control studies. There were no restrictions on dates of publication. Only English-language studies were included. The MEDLINE (Ovid) search strategy is provided in the protocol (Makleff et al., 2022).

### Screening process

Using Covidence (Veritas Health Innovation, 2017), we removed all duplicate articles. Next, three research team members screened articles for eligibility based on title, abstract and keyword. Two members of the research team independently assessed the full text of all remaining articles to determine eligibility for inclusion in the review. Articles were included if they fulfilled the following criteria: (a) were original research (excluded reviews, editorials and commentaries), (b) were written in English, (c) were conducted in a setting with people affected by drought/fire (fires with a natural cause, such as bushfires/wildfires), (d) included at least one adult participant from a vulnerable group, and (e) provided findings regarding mental health outcomes for vulnerable participants. There were no restrictions on the country of study or study design for original, peer-reviewed research studies; mixed methods studies were eligible for inclusion if they also fulfilled the inclusion criteria.

### Data extraction

We extracted the following data on study characteristics: year of data collection, year of fire/drought occurrence, year of publication, type of hazard, location of study, study design and description, study objectives/aims, vulnerable population characteristics, and key mental health findings for (a) the entire sample and (b) the vulnerable population in the study. We first summarized data separately based on type of hazard (fire/drought) and type of study (quantitative/qualitative). Next, we pooled and summarized the following data: type of hazard, study design, study location (country), year of publication and study sample size. We synthesized

additional findings relating to study methodology, types of vulnerabilities and mental health findings qualitatively.

### Quality assessment

All included studies were assessed for methodological quality using the Joanna Briggs Institute (JBI) critical appraisal tools, focusing on the extent to which the study had addressed the possibility of bias (Joanna Briggs Institute, 2020). Two members of the research team scored each paper; any discrepancies in scoring were discussed by the team to finalize the scoring. A numeric score was calculated for each paper in the review based on the total number of "yes" or "no"/"unclear" metrics of the JBI checklist (an "unclear" was assigned the same score as a "no"), as has been conducted in prior reviews (Bowring et al., 2016; Xu et al., 2017). Based on the JBI critical appraisal tools for each study design, qualitative studies were assessed on a ten-item scale, cohort studies on an eleven-item scale and cross-sectional studies on an eight-item scale. Quality assessment scores were compared across studies, and mean assessment scores with standard deviation were analyzed by study design. Following

Adalbert et al. (2021) and to provide comparisons, relative scores were depicted graphically to illustrate the percent value of each study relative to the others. These analyses elucidate the general quality of evidence of the existing literature and highlight the methodological strengths and weaknesses of the studies included in the systematic review.

## Results

### Screening of studies

Searches from all databases produced a total of 3,401 articles, and after removal of duplicates, 2,098 articles remained. With the removal of 2,066 articles after screening by title/abstract, 32 articles underwent full-text analysis, of which 18 were ultimately deemed eligible for inclusion in this review (see Figure 1).

### Study characteristics

Of the 18 studies included in this review, nine were in settings of drought and nine in settings of fire. Studies were conducted

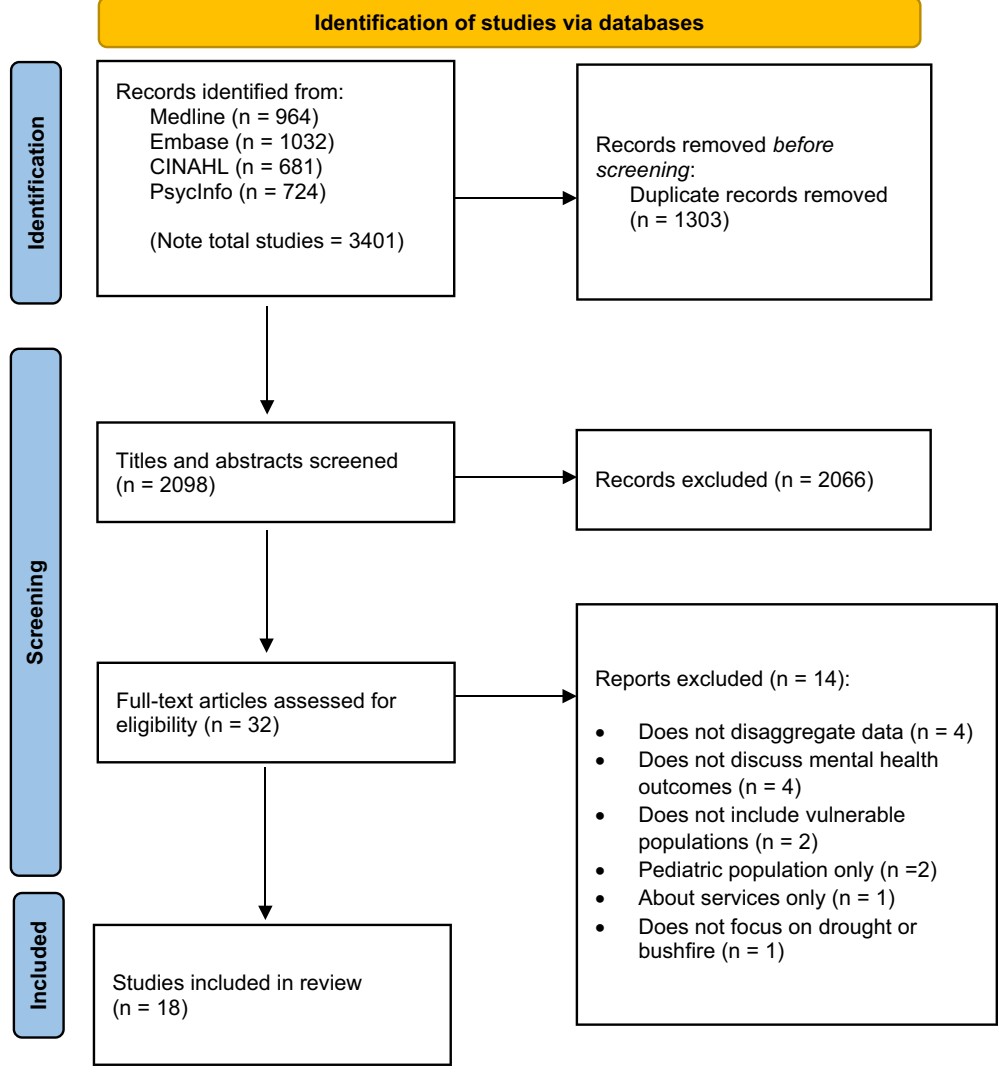

**Figure 1.** Process of screening articles for this scoping review based on the PRISMA 2020 flow diagram (Page et al., 2021).

between 2006 and 2022 in Australia (n = 11), the United States (n = 3), Greece (n = 2), Iran (n = 1) and Canada (n = 1). The number of drought- and fire-affected individuals by study ranged from 23 to 5,012 (unspecified in Hayati et al., 2010). The pooled total was 15,959 participants across all 18 studies. All but two studies in the review (Parslow et al., 2006; Scher and Ellwanger, 2009) included participants living in a rural or remote area, and ten of the studies focused exclusively on rural or remote residents. Other common vulnerabilities in the included papers were occupying a low socioeconomic position, experiencing chronic health conditions and/or mental health problems prior to the hazard event, having a low educational attainment, belonging to an ethnic minority group and being unemployed.

Measures of mental health varied by study. Qualitative studies, which used either individual interviews, surveys with open-ended questions, or focus groups, relied on participants' accounts of their experiences of different aspects of mental health. Most quantitative studies used self-report surveys and symptom checklists to assess mental health outcomes. Some studies used standardized measures such as the Trauma Screening Questionnaire (TSQ) (Brewin et al., 2002), Symptom Checklist 90-Revised (SCL-90-R) (Derogatis and Savitz, 1999), Kessler Psychological Distress Scale (K6), Kessler 10-L (K10) (Kessler et al., 2003), Impact of Events Scale-Revised (IES-R) (Weiss and Marmar, 1997) and the PTSD Symptoms Checklist (PCL-5) (Blevins et al., 2015) to determine the prevalence of symptoms of anxiety, distress, posttraumatic stress and anger. Other measures, including the Sense of Community Index (Chavis et al., 1986) and the Sense of Place (Shamai, 1991) scale, examined social support and community circumstances.

Data collection for included studies occurred at different time points relative to the hazard event. Studies occurred during the hazard event (n = 3; only for drought), in the same year as the event (n = 4), six months later (n = 3), one year later (n = 1), five years later (n = 1), in the year between multiple hazard events (n = 1), or they did not specify the time point relative to the hazard event (n = 4). One study (Carroll et al., 2022) collected data twice: two years after the fire and then six years after the event. Qualitative studies included in the review are listed in Table 1, quantitative studies in Table 2 and pooled characteristics for the studies are shown in Table 3.

## Quality assessments

Complete quality assessment critical appraisal checklist scorings are in Supplementary Tables S1–S3. Studies generally ranged from moderate to high quality overall. Qualitative studies (n = 6) had a mean of 73.3% of possible points (SD = 12.1; Range = 60–90%). Cross-sectional studies (n = 8) had a mean of 78.1% of possible points (SD = 17.4; Range = 50–100%). Cohort studies (n = 4) had a mean of 72.7% of possible points (SD = 19.61; Range = 45.4–90.9%). The most common methodological flaws identified across studies using the JBI criteria were inconsistencies identifying and addressing confounding factors, not reporting details about participants who were lost to follow-up (in cohort studies) and a lack of representation of participant perspectives (in qualitative studies). Quality assessments for all studies are depicted in Figure 2.

## Outcomes in fire-affected settings

Two qualitative studies (shown in Table 1), both in the United States, were conducted with participants who had experienced

fire in the 2017 Northern California Wildfires and the multiple fires between 2012–2020 in Okanogan, Washington. In one (Domínguez and Yeh, 2020), approximately one-third of the sample was deemed vulnerable due to one of the following factors: low income, being an ethnic minority, identifying as a sexual/gender minority, or having low educational attainment. In the other (Humphreys et al., 2022), all participants were vulnerable as the entire sample resided rurally, but further vulnerabilities were not clearly defined. In both studies, vulnerable individuals reported anger, cynicism, a perceived lack of support due to marginalization and an increased susceptibility to physical and mental health problems after experiencing fire.

Seven quantitative studies were conducted with participants who experienced fire (shown in Table 2). In three studies, all participants were vulnerable because they resided in a rural area. In four studies, between one-third and two-thirds of participants were classified as vulnerable due to unemployment, experiencing prior health or mental health conditions, belonging to an ethnic minority group (including being Indigenous), or having low educational attainment. Because the characteristics of vulnerability are not mutually exclusive, for these studies it was not possible to determine the total number of vulnerable participants in the study. People experiencing vulnerability often had mental health problems after the event, with prevalence estimates up to 36% for PTSD (Parslow et al., 2006; Austin et al., 2018; Belleville et al., 2021; Cowlishaw et al., 2021; Carroll et al., 2022), 10% for anger (Cowlishaw et al., 2021), 15% for anxiety (Scher and Ellwanger, 2009; Papanikolaou et al., 2011; Belleville et al., 2021) and 15% for depression (Scher and Ellwanger, 2009; Belleville et al., 2021); rates of psychosis and paranoia were noted to be high in one study, but numbers were unspecified (Papanikolaou et al., 2011). In a study in Greece, those with lower educational attainment had a higher risk of developing somatization symptoms, not further described (Papanikolaou et al., 2011). Indigenous people in Fort McMurray, Alberta, Canada, were shown to have more severe symptoms of mental illness, such as depression and anxiety, than those from other racial groups (Belleville et al., 2021).

Two of the quantitative studies examined mental health at different points in time. An Australian study (Carroll et al., 2022), which investigated the long-term impact of the bushfire-instigated Hazelwood mine fire, indicated that traumatic symptoms because of the fires not only lasted years after the event but also increased over time. The study found that younger participants (average age of 25 years, compared to groups with an average age of 45 and 65 years old) reported higher levels of ongoing distress in response to their exposure to the fire, even six years after the event, at the second round of data collection. These higher levels of stress likely occurred due to subsequent fire, as opposed to solely an increase in levels of stress occurring over time (Carroll et al., 2022). Further, media coverage about hazards, as well as similar fire- and smoke-related events locally, served as triggers elevating traumatic stress symptoms among participants. A Canadian study found that a prior history of mental health problems along with experiencing financial stress increased the odds of developing, or having more severe symptoms of, PTSD, depression, insomnia, anxiety and drug/alcohol dependency for those affected by fire (Belleville et al., 2021).

While most of the studies in the review did not compare between vulnerable mental health experiences and those in the general population, some fire-related studies employed other forms of comparison. Two studies in Greece compared those exposed and unexposed to the fires (Papanikolaou et al., 2011*a,b*). A prospective

**Table 1.** Qualitative study findings

| Study author, year, country | Objectives/aims | Year(s) of hazard event | Study description | Vulnerable groups in sample (not mutually exclusive) | Key mental health findings across study sample | Key mental health findings specific to vulnerable groups | Quality assessment score |
|---|---|---|---|---|---|---|---|
| | | | | Fire | | | |
| Dominguez & Yeh (2020), USA | To describe experiences with wildfires among impacted individuals and to generate insights for professionals supporting affected communities. | 2017 | *Sample:* 259 participants who experienced the 2017 Northern California Wildfires *Data collection:* Online survey including short answer and open-ended questions about wildfire experiences *Recruitment:* Convenience sample recruited through local organizations in fire-affected counties | • Most/all participants rural residents (number not specified) • 32% low income • 16% LGBTQ • 14% racial minorities • 4% low educational attainment | • Feelings of trauma, PTSD, anxiety, avoidance, loss, grief and panic were common. • Displacement was linked to feelings of loneliness, disconnection, separation and isolation. • Fires had long-lasting impacts. • There was an unmet need for support services. Clear pathways to these services are needed. • Caring for others added to feelings of being emotionally and mentally drained. | • Perception that support is unfairly distributed, and marginalized individuals lack access to support services. | 8/10 |
| Humphreys et al. (2022), USA | To describe the mental health and well-being impacts of extreme and persistent wildfire smoke events, how people have coped and opportunities to mitigate negative mental health impacts. | Persistent wildfire smoke events in summers of 2012, 2014, 2015, 2017, 2018, 2020 | *Sample:* 29 adult residents of Okanogan County (vacation destination in Washington, USA) *Data collection:* • Three focus groups with 13 community members • 16 key informant interviews with health and social service providers *Recruitment:* Participants recruited from community professional networks, social media, word of mouth and fliers posted throughout the community | • All focus group participants were rural residents • Other characteristics insufficiently reported and do not suggest vulnerabilities (primarily white, educated, employed) | • Feelings of anxiety, depression, isolation, lack of motivation, worry and stress were frequently mentioned. • Depression was described as related to social isolation. • There were lingering effects on mental health after the wildfires ended. • Key informants described a lack of accessible mental health services. • A range of stress reduction strategies at the community level were proposed for improving well-being. | • Some characteristics were described as increasing vulnerability or susceptibility to poor mental health, physical health and well-being: lower income, outdoor occupations, age (child or elderly), pre-existing health conditions (primarily respiratory conditions), pregnancy, housing insecurity or homelessness and social isolation. | 9/10 |
| | | | | *Drought* | | | |
| Hossain et al. (2008), Australia | To identify factors that influence the mental health of farmers and inform refinement of training materials to improve the well-being of this community. | Prolonged/ recurrent drought (timing not specified) | *Sample:* 23 farmers, health professionals, and organization representatives from rural Queensland *Data collection:* Three focus groups: • Farmers (n = 8) • Health professionals (n = 11) • Representatives from rural organizations (n = 4) *Recruitment:* Farmers identified through agricultural agencies, health | • All farmers (n = 8) were rural residents • Not stated if other participants are members of vulnerable groups | See column to right (all participants vulnerable) | • Ongoing drought was described as an additional source of pressure for farmers. • Participants commonly discussed having feelings of hopelessness, anxiety, depression and suicidal thoughts. • Farmers reported feeling isolated, hopeless and abandoned by society and the government. • Farmers reported stigma as an obstacle for mental health | 7/10 |

*(Continued)*

**Table 1.** (*Continued*)

| Study author, year, country | Objectives/aims | Year(s) of hazard event | Study description | Vulnerable groups in sample (not mutually exclusive) | Key mental health findings across study sample | Key mental health findings specific to vulnerable groups | Quality assessment score |
|---|---|---|---|---|---|---|---|
| | | | professionals nominated by their agencies and senior staff from rural organizations were invited to participate | | | care-seeking. Those accessing mental health services described a lack of continuity of care.<br>• Organizations supporting farmers would benefit from training about mental health needs of farmers.<br>• Health professionals expressed concerns about child exposure to dysfunctional behaviors, alcohol abuse and suicidal behaviors.<br>• Insufficient mental health support services in the community, and some farmers did not identify their need for support. | |
| Sartore et al. (2008), Australia | To explore how drought impacted on emotional and social well-being in a rural community and to inform strategies to support emotional well-being. | Prolonged/recurrent drought (timing not specified) | *Sample:* 39 rural community members in central-western New South Wales<br>*Data collection:* Focus groups:<br>• Female farmers (n = 10)<br>• Male farmers (n = 11)<br>• Local businesspeople (n = 6)<br>• Health and other support workers (n = 12)<br>*Recruitment:* Local key contacts invited community members to participate | • All participants were rural residents (n = 39)<br>• Other characteristics insufficiently reported to assess vulnerabilities | See column to right (all participants vulnerable) | • Drought-related changes in the environment and lack of rain brought about low mood, particularly for those who lived on a farm.<br>• Participants described feelings of guilt when watering plants in ongoing drought.<br>• Participants commonly reported feeling demoralized, worrying, negativity, fear for the future and uncertainty.<br>• Drought-related financial constraints brought about emotional challenges related to feelings of uncertainty.<br>• Participants felt resentful that their suffering related to drought was not understood.<br>• Solidarity, jokes, communal events and informal meetings helped participants. | 8/10 |
| Hayati et al. (2010), Iran | To understand how poor farmers cope with drought and the effects of government drought mitigation efforts | Prolonged/recurrent drought (timing not specified) | *Sample:* Drought-affected farmers in South Iran (sample size not specified)<br>*Data collection:* In-depth interviews<br>*Recruitment:* Local informants introduced researchers to drought- | • All participants were rural farmers (sample size not stated)<br>• Socioeconomic disadvantage (farmers classified as 'rich', 'moderate' and | See column to right (all participants vulnerable) | • Emotional and psychological effects of drought included feelings of depression, loss of confidence, general unhappiness, boredom and a questioning of faith.<br>• Loss of social interaction and increase in social isolation | 6/10 |

**Table 1.** (*Continued*)

| Study author, year, country | Objectives/aims | Year(s) of hazard event | Study description | Vulnerable groups in sample (not mutually exclusive) | Key mental health findings across study sample | Key mental health findings specific to vulnerable groups | Quality assessment score |
|---|---|---|---|---|---|---|---|
| | | | affected farmers; snowball sampling to recruit further participants | 'poor', but no information about these categories provided) | | amplified negative emotions, particularly depression.<br>• Poor farmers experienced negative psychological, social and economic consequences from drought conditions.<br>• Government interventions treated farmers homogenously, further disadvantaging low-income farmers.<br>• Economic challenges related to drought limited opportunities for farmers and their families, leading to depression and reduced self-confidence.<br>• Poor and moderately poor farmers mentioned reduced social interactions and related solitude leading to an increase in depression. | |
| Rigby et al. (2011), Australia | To describe how prolonged drought impacted rural Aboriginal communities and to discuss possible adaptive strategies to improve well-being. | Prolonged drought ("The Big Dry"), which was most severe in 2008 | *Sample:*<br>166 people from 27 communities requiring assistance during drought including Elders, key community members, representatives from Aboriginal organizations and other community organizations<br>*Data collection:*<br>Consultative community group forums across rural NSW in 2008<br>*Recruitment:*<br>Written invitations to attend the forums, follow-up invitations by telephone | • All participants Indigenous and/were or regional, rural, or remote residents | See column to right (all participants vulnerable) | • Connectedness to healthy land is essential for Aboriginal health and well-being; without connection to healthy land, health and well-being challenges were widespread.<br>• Drought affected social and emotional well-being and lowered self-esteem by promoting antisocial behavior, bringing shame to their culture.<br>• Mistrust, gossip and malicious behaviors were noted.<br>• Drought was linked to increased use of alcohol, leading to aggression, violence and suicidality.<br>• Loss of land and displacement from their lands led to feelings of grief and guilt.<br>• Despondency, despair, helplessness and hopelessness were common.<br>• Traditional family structure, culture and connection to place were harmed by climate impacts. | 6/10 |

**Table 2.** Quantitative study findings

| Study author, year, country | Objectives/aims | Year(s) of hazard event | Study description | Measures to assess mental health outcomes | Vulnerable groups in the sample | Key mental health findings across study sample | Key mental health findings specific to vulnerable groups | Quality assessment score |
|---|---|---|---|---|---|---|---|---|
| | | | | *Fire* | | | | |
| Parslow et al. (2006), Australia | To ascertain prevalence and risks for PTSD in young people following experiences of bushfire | 2003 | *Sample:* 2,085 people aged 20–24 years *Data collection:* PATH Through Life Project interviews in 1999 and follow-up interviews 3–18 months post bushfire about experiences and psychological symptoms and functioning *Recruitment:* Random selection of three age groups from ACT, NSW drawn from electoral rolls | • Goldberg's Depression and Anxiety Scores • Eysenck Personality Questionnaire Revised (EPQ-R) • Trauma Screening Questionnaire (TSQ) • Summed measures of social support obtained before the fire were used as indicators of social support during the fires. | Number unspecified. Groups include: • Socioeconomic disadvantage • Prior trauma • Poor mental health • Low educational attainment | • 36% experienced one or more trauma symptoms. • About 5% of all participants screened positive for PTSD. • 6% of those who experienced the fires firsthand displayed PTSD symptoms. | • Low levels of education, previous mental ill health, being evacuated from home or work during the fires, and grief due to losing a friend or relative to the fires or being injured in the fires, were associated with higher prevalence of PTSD symptoms. | 5/11 |
| Scher & Ellwanger (2009), USA | To examine effects of disaster-related cognitions, disaster experiences, non-PTSD psychopathology, and demographic characteristics on the development of depression, anxiety and somatic symptoms. | 2003 | *Sample:* 200 students affected by the 2003 southern California wildfires *Data collection:* Two questionnaires: one distributed 14–31 days after the fires; a second 6 months later *Recruitment:* Fliers posted throughout the university and in psychology classes | • Fire Impact Questionnaire (FIQ) • Posttraumatic Cognitions Inventory (PTCI) • Beck Anxiety Inventory (BAI) • Beck Depression Inventory-II (BDI-II) • Pennebaker Inventory of Limbic Languidness (PILL) symptoms | • 65% racial minorities | • Higher levels of negative thoughts related to fire (measured by the PTCI) were associated with increased symptoms of anxiety and depression. • Racial majority participants reported higher levels of anxiety symptoms compared to minority participants when negative cognitions related to the bushfire were high. | • Fire impact was associated with somatic symptoms. | 5/8 |
| Papanikolaou et al. (2011a), Greece | To investigate psychological functioning in those with severe exposure to wildfires, compared to those who did not experience the fire, and to identify risk factors for distress. | 2007 | *Sample:* 615 adult residents of the five prefectures declared disaster areas in 2007 fires: 353 in fire-affected group, 262 in control group *Data collection:* Questionnaires administered by interviewer | • Symptom Checklist 90-Revised (SCL-90-R) | • All participants were rural residents • 31.3% single/divorced/widowed • 27.5% low educational attainment | See column to right (all participants vulnerable) | • There were higher levels of somatization, depression, anxiety, obsessions and paranoia symptoms in people who were exposed to fire, than those who were not. • There were increased levels of somatization among people who lost property than those who did not. | 7/8 |

(*Continued*)

*Cambridge Prisms: Global Mental Health*

| Study author, year, country | Objectives/aims | Year(s) of hazard event | Study description | Measures to assess mental health outcomes | Vulnerable groups in the sample | Key mental health findings across study sample | Key mental health findings specific to vulnerable groups | Quality assessment score |
|---|---|---|---|---|---|---|---|---|
| | | | *Recruitment* No information provided | | | | • Losing a close relative to fire was associated with increased levels of paranoia. <br>• People exposed to fires without property damage, displayed a higher level of hostility compared those who had their property destroyed. | |
| Papanikolaou et al. (2011*b*), Greece | To understand the needs of those who experienced wildfires, their perceptions of major regional problems, and ways in which they can be supported. | 2007 | *Sample:* 800 participants: 409 adults living in villages affected by the 2007 wildfires in Greece, 391 comparable adults in areas unaffected by the fires *Data collection:* Questionnaire administered by interviewer *Recruitment* No information provided | • The Greek version of the Symptom Checklist – Revised (SCL-90-R) <br>• The Positive Symptom Distress Index (PSDI) <br>• The Positive Symptom Total (PST) | • Proportion of vulnerable participants not specified <br>• ~ 41% unemployed <br>• ~ 29% low educational attainment | • Exposure to fires was linked to higher levels of psychological distress. <br>• All participants reported better health status before the fires. <br>• Exposure to the fires was linked with higher levels of paranoid ideation, low mood and anxious feelings. | Not specified | 4/8 |
| Cowlishaw et al. (2021), Australia | To determine levels of anger and rates of psychological disorders after a bushfire, and to examine the association between anger and psychological issues. | 2009 | *Sample:* 796 residents from 25 rural/ regional communities across Victoria, Australia *Data collection:* Survey *Recruitment:* Participants from second wave of Beyond Bushfire Study (Gibbs et al., 2013; Bryant et al., 2017) were invited, with awareness-raising activities about the study including mail drops, phone calls and social media | • Dimensions of Anger Reactions Scale-5 (DAR-5) <br>• Four-item version of the PTSD Checklist (PCL-4) <br>• Patient Health Questionnaire (PHQ-9) <br>• Kessler Psychological Distress Scale (K6) | • All participants were rural residents and all experienced socioeconomic disadvantage | See column to right (all participants vulnerable) | • Anger problems were more prevalent in women than men. <br>• Participants affected by severe levels of bushfire had significant anger problems compared to participants with low to medium bushfire impact. <br>• Anger problems were associated with lower life satisfaction and a nearly 8-fold increase in suicidal ideation. <br>• Anger problems were linked to increase in hostile behaviors. | 8/8 |
| Belleville et al. (2021), Canada | To determine the prevalence of psychological morbidities 1 year | 2016 | *Sample:* 1,510 adult current or former Fort McMurray residents | • The PTSD Symptoms Checklist (PCL-5) <br>• The Insomnia Severity Index (ISI) | • ~ 30% racial minorities (6% First Nations) | • 38% of participants had a probable diagnosis of post-traumatic stress, | • Being part of the First Nations community was associated with more severe symptoms of mental illness | 6/8 |

**Table 2.** (*Continued*)

| Study author, year, country | Objectives/aims | Year(s) of hazard event | Study description | Measures to assess mental health outcomes | Vulnerable groups in the sample | Key mental health findings across study sample | Key mental health findings specific to vulnerable groups | Quality assessment score |
|---|---|---|---|---|---|---|---|---|
| | after evacuation from a major fire, and to determine disaster correlates to psychological disorders before, during and after the fire. | | who evacuated during the 2016 fires *Data collection:* Questionnaires administered over the phone *Recruitment:* Random digit sampling of mobile and home phone numbers | • The depression and anxiety subscales of the Patient Health Questionnaire (PHQ-9, GAD-7) <br>• The CAGE Substance Abuse Screening Tool | • ~6% socioeconomic disadvantage (unemployed/ on welfare) | depression, anxiety and substance abuse disorder. <br>• 29% of participants reported insomnia. <br>• Increased subjective levels of stress were reported after the fires. | than their non-Indigenous peers. <br>• Prior mental health problems were associated with an increased risk of mental illness and of financial stress. | |
| Carroll et al. (2022), Australia | To investigate the continued presence of, and changes in, the relationship between smoke exposure during the Hazelwood mine fire and subsequent event-related psychological distress. | 2014 | *Sample:* 709 adult residents of Morwell who were at least 18 years old at the time of the fire. *Data collection:* Two surveys evaluating posttraumatic distress, measured using the Impact of Events Scale-Revised (IES-R), three and six years after the mine fire. *Recruitment:* A weighted random sample of 1,512 were selected to participate from the 3,077 eligible adult residents of Morwell. | • The Impact of Events Scale-Revised (IES-R) | • All participants lived in the regional country town of Morwell. <br>• One-third of the cohort reported a mental health diagnosis prior to the mine fire (pre-2014). | See column to right (all participants vulnerable) | • Repeated data collection at 3 and 6 years revealed increases across all three posttraumatic distress symptom clusters, particularly intrusive symptoms. <br>• The follow-up survey coincided with the Black Summer bushfire season in south-eastern Australia and exposure to this new smoke event may have triggered distress sensitivities stemming from exposure to the earlier mine fire. | 9/11 |
| | | | | | *Drought* | | | |
| Stain et al. (2011), Australia | To understand the impact of prolonged drought on rural communities, and examine the role of social factors on psychological well-being. | 2010 (in broader context of recurring drought in the region) | *Sample:* 302 adult rural residents with high drought exposure *Data collection:* The Australian Rural Mental Health Study (ARMHS) survey *Recruitment:* Randomly selected from the Australian Electoral Roll (AER) | • Kessler Psychological Distress Scale —10 <br>• Worry about Drought Scale <br>• Short form of the EPI 2 item adverse life events scale | • All participants were rural residents, of which 23% were farmers <br>• 38% socioeconomic disadvantage (unemployed) <br>• 23% unmarried / widowed | See column to right (all participants vulnerable) | • Living on a farm or in a very remote area was linked to high levels of worry about the drought. <br>• 31% of participants were likely to have a mental illness as evidenced by the above clinical significance threshold on K10. <br>• Psychological distress was associated with a lack of well-being and loss of social connectedness. <br>• There was a higher likelihood of experiencing stress | 6/8 |

(*Continued*)

| Study author, year, country | Objectives/aims | Year(s) of hazard event | Study description | Measures to assess mental health outcomes | Vulnerable groups in the sample | Key mental health findings across study sample | Key mental health findings specific to vulnerable groups | Quality assessment score |
|---|---|---|---|---|---|---|---|---|
| | | | | | | | among those with higher neuroticism scores and lower for those with community/personal social support. | |
| Kelly et al. (2011), Australia | To determine factors that influence the psychological health of rural residents of drought-affected areas | Unspecified (recurring drought in region) | *Sample:* 2,639 adults in rural New South Wales *Data collection:* Baseline of the Australian Rural Mental Health Study (ARMHS) *Recruitment:* Stratified random sampling | • Kessler Psychological Distress Scale 10 (K-10) <br>• A subset of the Eysenck Personality Inventory- 12 (EPI-12) <br>• The list of life-threatening life experiences <br>• Sense of Community Index | • 100% rural residents (28% remote/very remote) <br>• 33% low educational attainment <br>• 5.6% socioeconomic disadvantage student/ carer/ home duties) | See column to right (all participants vulnerable) | • Worry about drought-affected well-being. <br>• Pre-disposition to neuroticism, recent adversity and personal lack of social support were associated with mental ill health. <br>• Having had three or more life-threatening events was associated with problems in personal social networks. <br>• Participants who lived in the most remote locations reported the lowest well-being and highest total K-10 scores, indicating the likelihood of a mental illness compared to those residing elsewhere. | 6/8 |
| Friel et al. (2014), Australia | To examine the associations between food security and mental health during drought. | 2003 | *Sample:* 5,012 respondents of Household, Income and Labour Dynamics in Australia (HLDA) survey *Data collection:* Secondary analysis, including of BOM data on rainfall and of data from Wave 7 of the HLDA survey *Recruitment:* No information provided | • Goldberg's Depression and Anxiety Scores <br>• Shortened form of the Eysenck Personality Questionnaire Revised (EPQ-R) <br>• Trauma Screening Questionnaire (TSQ) <br>• Summed measures of social support obtained before the fire as indicators of social support during the fires | • 18% rural residents <br>• 1.6% socioeconomic disadvantage | • Exposure to drought moderates the association between measures of food insecurity and psycholo-gical distress. <br>• Those with increased consumption of discretionary foods linked to higher levels of distress. | • Increased psychological distress was associated with food insecurity: those who missed meals and had financial stress reported moderate-high distress levels. <br>• Those exposed to constant and long drought in rural areas reported having more distress compared to those in any other category. | 8/8 |
| Powers et al. (2015), Australia | To determine the impact of drought on the psychological health of women and vulnerable populations in rural settings. | 2007 (study included data from 1996, 1998, 2001, 2004, 2007) | *Sample:* 6,664 women, of which 492 were affected by drought and 6,172 not affected by drought. Those in rural/ remote areas oversampled for representation | • Mental Health Index of the validated Medical Outcomes Study Short Form 36 (SF36) | Overall sample: <br>• 48.6% low educational attainment <br>• 42.5% health condition <br>• 28.6% socioeconomic disadvantage | • No association was found between drought and women's mental health. <br>• Mental health help-seeking behaviors did not vary by drought conditions (2001–2007). | • Women with difficulties managing on their available income and those with poor mental health in 1996 had significantly poorer Mental Health Index scores compared with the general population of women. <br>• Less educated women and those with poor mental | 10/11 |

(*Continued*)

**Table 2.** (Continued)

| Study author, year, country | Objectives/aims | Year(s) of hazard event | Study description | Measures to assess mental health outcomes | Vulnerable groups in the sample | Key mental health findings across study sample | Key mental health findings specific to vulnerable groups | Quality assessment score |
|---|---|---|---|---|---|---|---|---|
| | | | *Data collection:* Surveys mailed as part of the Australian Longitudinal Study on Women's Health *Recruitment:* Random selection of rural women across Australia | | Participants exposed to drought:<br>• 16.7% remote/ very remote<br>• 12.9% socioe-conomic disad-vantage<br>• 2.9% widowed | | health in 1996 that experi-enced drought had slightly improved mental health over time. | |
| Austin et al. (2018), Australia | To examine personal and community drought-related stress among farmers and any influence of socio-demographic and community factors on these types of stress. | 1997–2010 | *Sample:* Rural farmers at three timepoints:<br>• Baseline: n = 664<br>• 3-year follow-up: n = 279<br>• 5-year follow-up: n = 235<br>*Data collection:* Through the Australian Rural Mental Health Study (ARMHS) *Recruitment:* Stratified random sample in non-Metropolitan New South Wales via the Australian Electoral Roll | • The Kessler Psychological Distress Scale (K10)<br>• Personal drought-related stress (PDS)<br>• Community drought-related stress (CDS)<br>• Eysenck Personality Ques-tionnaire (short form) (EPQN)<br>• List of Threatening Experi-ences questionnaire<br>• Sense of community index<br>• Sense of Place Scale | • All participants rural residents (10% remote; 90% regional)<br>• 36% socioeco-nomic disad-vantage<br>• 27.4% low edu-cational attain-ment<br>• 19% aged 65+<br>• 17.5% separ-ated/ widowed/ unmarried | See column to right (all participants vulnerable) | • Drought-related stress con-tributed to general psycho-logical distress.<br>• Socio-demographic and community factors including age, unemployment and remoteness influenced stress levels.<br>• Men experienced greater psychological distress com-pared to women.<br>• Higher trait neuroticism was associated with higher levels of psychological distress.<br>• Good mental health and relationships were associ-ated with lower levels of general distress.<br>• Younger participants (under 35), those experiencing greater financial insecurity, and those living in outer regional and remote areas reported higher levels of psychological distress.<br>• Those who both lived and worked on a farm had a higher incidence of drought-related stress than those who either lived or worked on a farm.<br>• Drought-related stress was higher with increased remoteness of location.<br>• General psychological dis-tress was not influenced by remoteness.<br>• Experience of 4–6 adverse life events was associated with greater personal, com-munity and drought-related stress and general psycho-logical distress. | 8/11 |

**Table 3.** Pooled study findings (total studies = 18)

| Study characteristic | Number of studies (%) |
|---|---|
| *Type of hazard* | |
| Drought | 9 (50) |
| Bushfire | 9 (50) |
| *Study Design* | |
| Qualitative research | 6 (33.3) |
| Cross-sectional | 8 (44.4) |
| Cohort design | 4 (22.2) |
| *Country* | |
| Australia | 11 (61.1) |
| Canada | 1 (5.5) |
| Greece | 2 (11.1) |
| Iran | 1 (5.5) |
| United States of America | 3 (16.6) |
| *Year of publication* | |
| Before 2010 | 4 (22.2) |
| 2010–2016 | 8 (44.4) |
| After 2016 | 6 (33.3) |
| *Link between MH outcome and vulnerability* | |
| Explicit | 13 (72.2) |
| Implicit | 4 (22.2) |
| Unclear/none | 1 (5.5) |
| *Range of number of affected individuals by study* | 23–5,012 |
| *Total affected individuals* | 15,959 |

cohort study in Australia focused on the mental health impacts immediately after a bushfire and compared these with the same participants four years later (Parslow et al., 2006). Another Australian study stratified data based on age and the level of exposure to fire among participants in the Hazelwood mine fire study (Carroll et al., 2022).

### Outcomes in drought-affected settings

Across the four qualitative studies conducted in settings affected by drought (shown in Table 1), all participants were deemed to be vulnerable as they resided in rural settings. Other aspects of vulnerability in these studies included low socioeconomic position and being Indigenous. Mental health problems such as depression, addiction, anxiety and suicidal thoughts and behavior were described by participants in these studies, who discussed the loss of community, and resultant decline in social interaction, as contributing to their mental health problems. Lost income, alongside a need to work longer hours, and in addition to drought-related worry, detracted from well-being; participants described a lack of accessible mental health services as a barrier to recovery.

Five quantitative studies, all conducted in Australia, focused on drought (shown in Table 2). The proportion of vulnerable participants in these studies ranged from 18% to 100%, with one study not reporting the number of participants classified as vulnerable (Parslow et al., 2006). The most frequent types of vulnerability in these studies were living in a rural/remote area, having a low level of education and being unemployed or having a low income. Worry, distress and overall poor mental health (which was not further defined in one study (Powers et al., 2015) were shown to be higher among certain vulnerable populations. For example, for drought-affected participants, living in very remote areas was associated with each of the following factors: a high likelihood of having symptoms of mental illness (Austin et al., 2018), low levels of well-being and high worry (Kelly et al., 2011., 2010; Stain et al., 2011). Drought-affected individuals experiencing financial/food insecurities and social isolation due to their remote or regional locations tended to have poor mental health outcomes. For example, Friel et al. (2014) found that higher levels of psychological distress were significantly associated with experiencing financial stress and food insecurity. Another study found that unemployed individuals were at four times higher risk to develop mental illness compared to their employed peers (Austin et al., 2018).

### Discussion

This systematic review contributes to our understanding of the mental health outcomes and experiences of vulnerable groups affected by natural hazards, specifically fire and drought. It provides evidence, primarily from high-income countries, that vulnerable individuals affected by drought or fires are more likely to experience anxiety, depression and general distress than less vulnerable groups. Common experiences reported by vulnerable individuals affected by drought included worry and, at worst, suicidality. Those affected by fire reported symptoms of PTSD and anger and there was some evidence of increased risk of psychosis. Time scale is a possible explanation for the differences in mental health outcomes after drought compared to bushfire. Drought generally occurs over long time periods, with prolonged stressors that can gradually contribute to mental health consequences such as suicidality (Padhy et al., 2015), while bushfires are rapid-onset events and may trigger different mental health consequences through different mechanisms (Askland et al., 2022; Zhang et al., 2022).

Notably, mental health problems arising after these natural hazards were described as exacerbating already existing mental health, physical health and socioeconomic challenges. For example, poor mental health prior to the disaster event increased the odds of developing financial problems and mental illness post-hazard (Parslow et al., 2006; Belleville et al., 2021; Carroll et al., 2022; Humphreys et al., 2022). Similarly, those who had PTSD prior to hazard events tended to have worse mental health outcomes after the event (Domínguez and Yeh, 2020). Other research can help understand why vulnerabilities such as poverty or hazard experiences can exacerbate poor mental health. Most mental health problems, including depression and suicidality, are underpinned psychologically by experiences of being entrapped and feeling powerless and unable to escape; this may be common in some forms of vulnerability, such as poverty and interpersonal violence (Fisher et al., 2020). These feelings can be made worse if the situation is intrinsically humiliating, as in the case of being marginalized or discriminated against, rejected, having a sense of subjective incompetence compared to others, or having fewer capabilities or resources (Fisher et al., 2020). Further, there is evidence that helps understand why natural hazards may exacerbate poor mental health. Direct trauma and physical danger, as well as indirect damage to personal environment, livelihood and

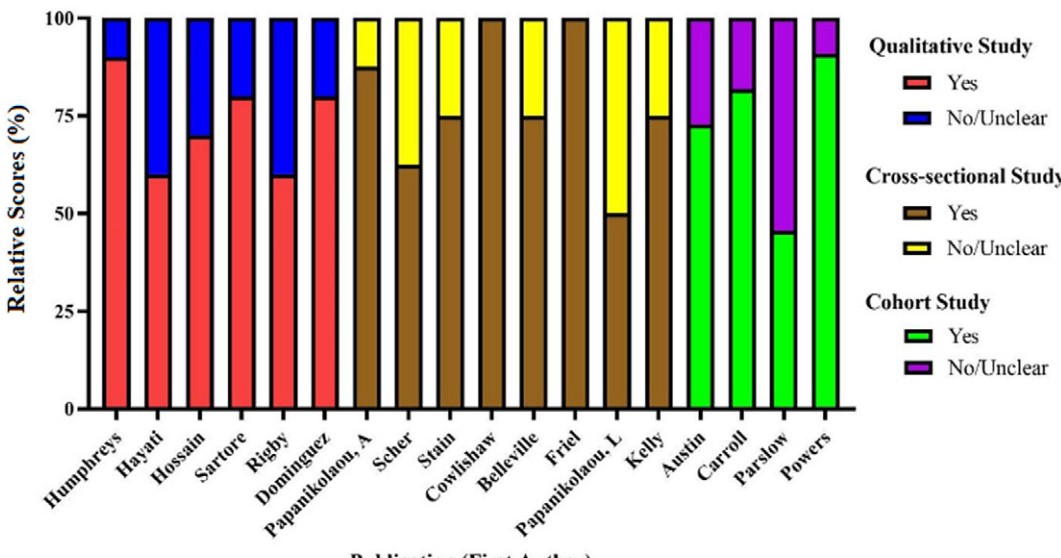

**Figure 2.** Quality assessment scores by study design.

property, have been identified as aspects that may exacerbate mental health conditions in the case of bushfires (Zhang et al., 2022). Factors that may exacerbate physical and emotional challenges after drought include the emergence of air pollution, a loss of access to fresh water and compromised agricultural production with concurrent damage to peoples' livelihoods (Vins et al., 2015).

The most common type of vulnerability in the included studies was residing in a rural setting. This reflects that fire and drought are more common in rural and remote than urban settings, where farmers and Indigenous and First Nations communities, among others, often already face socioeconomic challenges. These communities rely on drought- or fire-affected lands for their livelihoods and their mental well-being. Together, the findings across studies indicate that the mental health needs of the most vulnerable members in a community warrant specific consideration following a natural hazard. This review highlights a lack of examination in the current literature of the intersectionality of vulnerability factors and the potential compounding effect that these may have on mental health after natural hazard exposure. As social characteristics cannot be understood separately from each other (Weldon, 2008), an intersectional lens helps examine how overlapping forms of marginalization and discrimination can impact the lives of individuals (Victoria Government, 2021). Intersectionality encompasses the layering of individual characteristics such as age, location, ethnicity and gender (Walker et al., 2021). Historically used in gender and racial justice movements, intersectionality has also been applied to understanding how climate change and climate hazards can exacerbate existing inequalities (Kaijser and Kronsell, 2014; Thompson-Hall et al., 2016; Walker et al., 2021). The dearth of studies in this review that explicitly examine vulnerability, let alone the intersection between different types of vulnerability, emphasizes the need to adopt an intersectional lens when studying post-hazard mental health outcomes in the future. This is important because overlapping vulnerabilities across the lifespan influence not only the hazard experience but also the ability to recover and rebuild from the hazard.

The findings have implications for psychologically informed responses across the spectrum from promotion of mental health and prevention of mental health problems to early intervention and treatment. These include non-health-sector and health-sector actions and the need for these to be culturally safe and explicitly inclusive of members of vulnerable populations. The findings also have implications for community-centred hazard response. During and immediately after a hazard, it is important to consider strategies to promote inclusive community cohesion and peer-to-peer recovery activities to address immediate safety and survival needs (Chang, 2010; Ludin et al., 2019). Clear communication and transparency from local and state authorities about emergency and recovery services and resources and inclusive recovery planning, in which diverse community members are represented, is an approach that can promote trust and reduce frustration (Rosenberg et al., 2022).

Loss of income-generating work and property, including homes, farm infrastructure, stock and crops, contributes to despondency and hopelessness after fire and drought. Programs and resources to address the urgent priorities of food and financial insecurity, and emergency housing, tailored to ensure that vulnerable populations have equity of access, are vital. Provision of job-acquisition support programs and other training programs to improve livelihoods will also remain valuable. An example of such a program is the Catchment Management Authority Drought Employment in Victoria, Australia, where those affected by drought are employed in public-good and environmental projects while concurrently developing new skills that will help them become employable in other spheres (Victoria State Government, 2021).

A strong case has been articulated for policies that include social and financial support for services aimed at reducing health inequities and structural vulnerabilities throughout the various phases of a disaster – from pre-disaster planning phase to the chronic post-traumatic reestablishment phase (McFarlane and Williams, 2012, Finucane et al., 2020). Social policies to reduce inequities, for example by improving income, can address some of the underlying contributors to poor mental health and thus have the potential to indirectly improve mental health. For example, a longitudinal study of flood survivors in Germany showed that financial support, alongside supportive counseling, was associated with lower levels of mental health strain among vulnerable individuals (Daniel and Michaela, 2021).

In terms of health sector approaches, the findings suggest that strategies to improve mental health need to consider the structural barriers impeding access to mental health services such as stigma, lack of affordability and limited availability of service providers, particularly in rural areas, as has been suggested elsewhere (Cosgrave et al., 2015; Morgan et al., 2021). Inclusion of primary care practitioners in identifying people with poor mental health and delivering mental health services, particularly in rural and remote areas, is one approach that can be considered to increase access to mental health support (McFarlane and Williams, 2012).

Longer-term investments in supporting mental health are also recommended. Drought and fires can have lasting negative impacts on vulnerable populations and repeated hazard events can amplify these experiences (Vins et al., 2015; Cianconi et al., 2020). Adverse mental health impacts are evident years after a hazard (Raker et al., 2019) and long-term support services are needed that include mental health support for hazard-affected communities, particularly for vulnerable members of these communities (Wilson-Genderson et al., 2018). To enhance relevance and acceptability, such services should be co-designed with communities and consider cultural safety.

Many people seek psychologically informed practical assistance rather than specific psychological services, and some identify a need for crisis counseling services focused on mental health (Jogia et al., 2014). However, mental health services alone will be insufficient. It is critical that investment also be placed into programs and interventions that prioritize mental health promotion, as well as those which seek to address the underlying risk factors for mental health problems, such as financial insecurity, domestic violence and discrimination (Oram et al., 2016; Vargas et al., 2020; Virgolino et al., 2022).

The strengths of this review are that the search strategy was designed by a specialist information analyst, the protocol was pre-published (Makleff et al., 2022) and it followed standard guidelines. In addition, it included a quality assessment process that allows for an interpretation of the findings taking study quality into account. We acknowledge the limitation that the search was restricted to studies published in English and relevant studies published in other languages might have been missed. Further, our search terms may have missed relevant papers that examine indirect aspects of drought or fire experiences. Nevertheless, we believe that the strengths of the study outweigh its limitations and that it provides an accurate account of the state of knowledge in this field.

There are methodological strengths and limitations in this body of evidence. First, a subset of included studies only had a partial focus on vulnerable individuals and had limited comparison of the mental health experiences of vulnerable individuals and the general population. Second, the heterogeneity in measures precluded meta-analysis and we are unable to estimate the prevalence of mental health outcomes in vulnerable populations after drought or fire with precision. Third, quality ratings varied among studies corresponding to the various study designs, methods of recruitment, use of instruments and efforts to minimize possible biases. One of the main detractors of quality identified through the appraisal process was a lack of consideration of potential confounding variables. Last, while this review did include findings from different cultural and national contexts, most studies were conducted in Australia and other high-income countries. It is possible that some of the findings are not relevant to low- and middle-income countries, which may have fewer resources to support affected populations. In terms of methodological strengths, while vulnerable groups and the general population were not compared in any papers in the review, four studies (Parslow et al., 2006; Papanikolaou et al., 2011*a,b*; Carroll et al., 2022) did use other forms of comparison (exposed vs. unexposed, over time and by level of exposure) to examine mental health outcomes.

To strengthen the body of evidence in this burgeoning field, future research could focus on intersectional experiences of vulnerability and examine potential confounders that may have influenced mental health experiences. In addition, studies should aim, as appropriate for their research questions, to incorporate standardized measures that have been tested for their reliability and validity to allow for comparison of data beyond the particular study (Boynton and Greenhalgh, 2004; Boateng et al., 2018). Key factors to consider in the selection of measures would be formal validation against a gold standard diagnostic measure, comprehensibility for people of diverse literacies and ideally having been used in equivalent studies (Boynton and Greenhalgh, 2004; Boateng et al., 2018). It is beyond the scope of this paper to review individual measures though this would be a valuable area of future research activity.

Our review adds to evidence from prior systematic and scoping reviews about the mental health of people who have experienced a natural hazard (Laugharne et al., 2011; Finlay et al., 2012; Vins et al., 2015) by synthesizing the available evidence about the mental health of vulnerable communities who have experienced fire and drought. Based on our findings, this focus on vulnerability has relevance for the mental health of farming, rural, and Indigenous and First Nations communities that depend on the land for their livelihoods, who live in settings that are experiencing catastrophic fires and extended drought more frequently.

## Conclusion

This systematic review contributes to a more comprehensive understanding of the mental health consequences of natural hazards among vulnerable communities. The evidence indicates that many members of vulnerable groups experience mental health problems after exposure to drought and fire, including PTSD, depression, anxiety, suicidality, overuse of alcohol and anger. We found that limited access to mental health services, isolation and loss of community and income were drivers of mental health problems in these communities.

This review highlights the importance of improving the evidence base about mental health in vulnerable communities affected by natural hazards by including standardized measures and comparison groups. Further, there is a gap in studies that examine the role of intersectional vulnerabilities and systematically disaggregate data to allow for analysis of the particular mental health experiences of vulnerable communities after disaster. Future studies that draw on these approaches to examine the mental health effects of drought and fire on vulnerable individuals will help ensure that programs are informed by an understanding of the unique needs of these communities.

Findings have relevance for post-disaster efforts and can be used to inform policies and programs to help vulnerable groups build their resilience against hazards and prepare for, respond to, and recover from disasters. In conclusion, the mental health of vulnerable individuals and communities recovering from natural hazards must be considered and addressed as part of holistic recovery efforts aiming to improve health and well-being in the context of structural disadvantage.

**Open peer review.**  To view the open peer review materials for this article, please visit http://doi.org/10.1017/gmh.2023.13.

**Supplementary material.**  The supplementary material for this article can be found at https://doi.org/10.1017/gmh.2023.13.

**Data availability statement.**  Data available within the article or its supplementary materials.

**Acknowledgements.**  We acknowledge the valuable contribution to this research of the Monash University–led Fire to Flourish (2021-2026) program team and partners.

**Author contribution.**    S.M., R.N.K., L.R., J.F. conceptualized the study; the research strategy for the Fire to Flourish project overall was supported by R.W. L.R. developed the search strategy and conducted the search. K.V., S.M., K.V., R.N.K. contributed equally to data extraction and quality assessment. K.V., S.M., R.N.K. contributed equally to writing the paper, with further contributions from L.R., J.W., R.W., J.F. R.V. developed all tables and figures, with further contributions from S.M., R.W., J.W., J.F. All authors reviewed and approved the final manuscript.

**Financial support.**    Cornerstone funding for Fire to Flourish, which supported this review, is provided by the Paul Ramsay Foundation and Metal Manufactures Pty Ltd., with additional philanthropic support from the Lowy Foundation. The funders had no influence on any aspect of the protocol development or the implementation of the review. JF is supported by the Finkel Professorial Fellowship, which receives funding from the Finkel Family Foundation.

**Competing interest.**    The authors declare none.

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
