## [Reviewer Report]

2 November 2022

Global Mental Health

Editorial Director

Dear Editorial Committee,

We are pleased to submit the attached manuscript entitled “Impact of experiencing a drought or bushfire on the mental health of vulnerable groups: a scoping review” for consideration for publication. The objective of this scoping review is to describe the mental health outcomes of vulnerable populations after droughts and bushfires/wildfires and identify gaps in the literature. This falls within the journal’s remit to contribute to the emerging field of global mental health.

We confirm that neither the manuscript nor any parts of its content are currently under consideration or published in another journal. All authors have approved the manuscript and agree with its submission to Global Mental Health: Cambridge Prisms. None of the authors report a conflict of interest.

The word count is 4668 words excluding abstract, references, and supplemental tables. 

Best wishes,

Professor Jane Fisher

JANE FISHER AO, BSc (Hons), PhD, MAPS, FCCLP, FCHP

Finkel Professor of Global Health

Head, Division of Social Sciences

Immediate Past President International Marcé Society for Perinatal Mental Health

Global and Women’s Health

Public Health and Preventive Medicine

Monash University

Level 4, 553 St Kilda Rd

Melbourne VIC 3004

Australia

E: jane.fisher@monash.edu

---

## [Reviewer Report]

*Comments to Author*: Thank you for the opportunity to review this really interesting and timely scoping review which I think makes a valuable contribution to the literature.

I found this to be a really well written paper, and so only have a few minor comments which you may want to consider. The only comments below where I think minor revision is definitely required are those relating to the numbers on the PRISMA flow chart and a few corrections in the reference list. 

Line 2 - Later on you drop the hyphen between post and traumatic, in line with current APA preferences, so I would drop the hyphen here

Line 13 - the text uses both British and US spelling throughout - I note the journal allows either, but I would suggest being consistent

Line 43 – it is not clear what “drought losses” is referring to. Looking at the source article, I think this should be economic losses.

Line 104 – the citation for Clarivate looks odd - my guess is the company name is being interpreted as initials.

Line 136 – you say that “scores were depicted graphically”. Should this be relative scores given you depict percentages?

Line 224 – As this is an open review, I understand that my details will also be public, so I should note that the Carroll et al. paper you are referring to here is one that I was lead author on. As a result, I can comment directly on this particular paper, and am wondering if it might be worth adding here that this second round increase in distress was likely the result of a subsequent fire event rather than a general increase in distress over time. This interpretation of the data was made clear in this paper.

Line 389 – You highlight here and in the abstract, the importance of standardised measures, and note earlier that there was variation in the measures used. I am wondering if there is an opportunity to suggest one or more measures that might be used in future research to increase comparability between studies?

Line 483 – as noted earlier, the reference to Covidence appears to be formatting incorrectly

Line 544 – Tierney reference appears incomplete

Line 551 – Reference also appears incomplete

Line 558 – Should there be a date this was accessed

Lines 582 and 583 – these WHO citations appear incomplete – are they webpages?

Figure 1 (page 47) – there are a few points where the numbers on the PRISMA flow chart don’t add up.

- 3401 total studies minus 1303 duplicates should tally to 2098, rather than 2025, so there seems to be another 73 articles that were excluded

- Similarly, 2025 titles and abstracts minus 1551 exclusions leaves 474 articles rather than 32, so there are 442 articles that need to be accounted for.

Figure 2 (page 48) – the y-axis might be more appropriately labelled as “Relative score”?

Supplementary material

- I assume the formatting of the supplementary information is at the authors' discretion - so this may be better presented in landscape and with less than double spacing to fit each table and associated descriptors on the same page? 1.5 spacing appears to work well.

- The heading for the left hand columns isn’t quite correct. Suggest changing it to “First Author (Year) – Country” on all three tables.

- Following each table, you may want to add a carriage return and a heading like “Assessment criteria” before listing the 8-11 items.

---

## [Reviewer Report]

*Comments to Author*: The scoping review is relevant in the context of climate change and planetary health and its increasing effects on the mental health of populations. The methods used in conducting a scoping review are also sound with appropriate guidelines and checklists employed.

I had a few comments that could be considered by the authors. They do not need to be agree upon and can rebut these.

1. The word impact often refers to causality in research terminology and is loosely used to mean an association/ correlation. In this context and the studies reviewed, would you like to consider using the word impact in the title of the manuscript.

2. Why were a scoping review chosen over other literature review/ synthesis methods? Has this been mentioned or been alluded to in the manuscript?

3. Why was bushfire and droughts chosen over other natural hazards, like cyclones or floods? What was the rationale of choosing these two specific natural hazards? Was there an a priori decision or an inductive process that went into this or was it an outcome as a part of another study. A little more details on this would strengthen the quality of the manuscript.

4. The manuscript mentions the synthesis of global literature in several places. In the light of the review including literature from only five countries and excluding non-English language studies, would the authors still like to use the term global in the manuscript?

5. Were there any mixed methods studies identified during the search/ screening of literature, and if so, how were they dealt with and categorised?

6. The results of the review bring out the differences in the mental health effects of bushfire and drought which is mentioned in the discussion section of the manuscript. However, why is there a difference in the mental health symptoms of fire and drought can be discussed a little more with reference to existing literature and what could be the possible reasons for these differences? 

7. In mental health/ illness literature, natural hazards act as life events in precipitating and perpetuating mental illness. Can this be discussed a bit more which comes out in the findings of the review.

---

## [Reviewer Report]

*Comments to Author*: This is a well written review and I agree it could be called a systematic review as indicated by one reviewer and am wondering the reason for not stating so. Besides responding to the comments made by the reviewers it would be good to get more discussions around the qualitative scores and the implications of such on the results.

---

## [Reviewer Report]

Dear Editors,

Thank you for giving us the opportunity to submit a revised draft of the manuscript currently titled “Mental health outcomes of vulnerable groups experiencing a drought or bushfire: a systematic review” to Cambridge Prisms: Global Mental Health. We appreciate the time and effort that you and the reviewers have dedicated to providing valuable feedback to strengthen our manuscript. We are grateful to the reviewers for their insightful comments. Please find our response to the reviewers as well as the changes made in the manuscript based on the reviewer’s comments in the table below. We have also attached a revised version of the manuscript with tracked changes. 

We also present to you a draft of the visual abstract, which includes two images. However, we acknowledge that creating a visual abstract is not our area of expertise. We are keen to co-produce a visual abstract if the journal is able to support us. We look forward to hearing from you and moving this manuscript further. 

Our detailed responses have been uploaded as a PDF to the portal. 

Sincerely,

Dr. Shelly Makleff on behalf of the author team

---

## [Reviewer Report]

*Comments to Author*: Thank you for the opportunity to review this revised paper which I think has benefited from the review process. I agree with the decision to change it to a systematic review, which better fits the paper. I only have a couple of very minor additional comments for your consideration. Well done on making a great contribution to the research literature.

line 116 - The citation for covidence still looks incorrect - it refers to Innovation VH instead of Veritas Health Innovation - this is because your citation manager is treating the company name as a person and so the first two terms are presented as initials. If you are using endnote you can address this by adding a comma after the organisation name in the endnote author field, which forces the full name to be cited.

Line 255-257 - this new paragraph is a good addition. While the current review has focussed on the Carroll et al. 2022 paper, which looked at differences within the exposed community across two survey rounds, it might be worth noting that an earlier paper on the first survey round results did involve a comparison with an unexposed community (Sale) - Broder, J. C., Gao, C. X., Campbell, T. C. H., Berger, E., Maybery, D., McFarlane, A., Tsoutsoulis, J., Ikin, J. F., Abramson, M. J., Sim, M. R., Walker, J., A, L., & Carroll, M. (2020). The factors associated with distress following exposure to smoke from an extended coal mine fire Environmental Pollution 266, 115131

line 622 - tierney reference still looks incomplete - looks like it should be Tierney, Kathleen. “Social Inequality, Hazards, and Disasters”. On Risk and Disaster: Lessons from Hurricane Katrina, edited by Ronald J. Daniels, Donald F. Kettl and Howard Kunreuther, Philadelphia: University of Pennsylvania Press, 2006, pp. 109-128. https://doi.org/10.9783/9780812205473.109

line 623 - UNEP reference is still incomplete as it ends with a double period and no information on what type of reference this is - google suggests it is .a media release so the reference should include [Press Release]followed by web address. Alternatively, it may be better to cite the underlying report rather than the media release - looks like it is available at https://www.unep.org/resources/report/spreading-wildfire-rising-threat-extraordinary-landscape-fires

---

## [Reviewer Report]

Dear editorial team, 

We are delighted to share with you a finalised version of our paper GMH-22-0244.R1 entitled “Mental health of vulnerable groups experiencing a drought or bushfire: a systematic review”. This draft takes into account all reviewer comments, as detailed in our response to reviewers. Further, per communication with your team, we have decided not to include a graphical abstract. 

We look forward to your feedback on this submitted draft. Please contact us with any further queries.

Sincerely,

Shelly Makleff and Professor Jane Fisher on behalf of the co-author team